# Using `SimTeEx` to simplify polynomial expressions with tensors

Renato M. Fonseca

High Energy Theory Group
Departamento de Física Teórica y del Cosmos,
Universidad de Granada, E–18071 Granada, Spain

Email: renatofonseca@ugr.es

**Abstract**

Computations with tensors are ubiquitous in fundamental physics, and so is the usage of Einstein's dummy index convention for the contraction of indices. For instance, $T_{ia}U_{aj}$ is readily recognized as the same as $T_{ib}U_{bj}$, but a computer does not know that `T[i,a]U[a,j]` is equal to `T[i,b]U[b,j]`. Furthermore, tensors may have symmetries which can be used to simply expressions: if $U_{ij}$ is antisymmetric, then $\alpha T_{ia}U_{aj} + \beta T_{ib}U_{jb} = (\alpha - \beta) T_{ia}U_{aj}$. The fact that tensors can have elaborate symmetries, together with the problem of dummy indices, makes it complicated to simplify polynomial expressions with tensors. In this work I will present an algorithm for doing so, which was implemented in the Mathematica package `SimTeEx` (*Simplify Tensor Expressions*). It can handle any kind of tensor symmetry.

## 1. Introduction

In particle physics as well and in general relativity one often has to deal with expressions involving tensors, such as the Riemann tensor or the Wilson coefficients of an effective field theory. The indices of such tensors appear so often contracted that implicit summation of repeated indices [1] is by now second nature to researchers in these fields.

Index contractions and the potential symmetries under exchange of indices can make it non-trivial to simplify expressions involving polynomials of tensors. For example the symmetries of the Riemann tensor,

$$R_{ijkl} = -R_{jikl}\,, R_{ijkl} = -R_{ijlk} \text{ and } R_{ijkl} + R_{iklj} + R_{iljk} = 0\,, \tag{1}$$

imply that [2]

$$R_{pqrs}R_{ptru}R_{tvqw}R_{uvsw} - R_{pqrs}R_{pqtu}R_{rvtw}R_{svuw}$$
$$-R_{mnab}R_{npbc}R_{mscd}R_{spda} + \frac{1}{4}R_{mnab}R_{psba}R_{mpcd}R_{nsdc} = 0\,. \tag{2}$$

Likewise, in the Standard Model effective field theory (SMEFT), at dimension 6, one encounters the operator [3–5]

$$\mathcal{O}_{ijkl} = \epsilon_{\alpha\beta\gamma}\epsilon_{nm}\epsilon_{pq}\left(Q_{i,\alpha n}^T C Q_{j,\beta p}\right)\left(Q_{k,\gamma q}^T C L_{l,m}\right) \tag{3}$$

where the external indices $(ijkl)$ label the fields' flavor. The remaining subscripts are $SU(3)$ and $SU(2)$ indices, which are unimportant for the present discussion. Crucially, $\mathcal{O}_{ijkl}$ is a tensor with the non-trivial symmetry [5]

$$\mathcal{O}_{ijkl} + \mathcal{O}_{jikl} - \mathcal{O}_{kijl} - \mathcal{O}_{kjil} = 0\,. \tag{4}$$

It follows that the Wilson coefficient $\kappa_{ijkl}$ which contracts with $\mathcal{O}_{ijkl}$ in the SMEFT Lagrangian is the most general tensor obeying the slightly different relation

$$\kappa_{ijkl} + \kappa_{jikl} - \kappa_{jkil} - \kappa_{kjil} = 0 \tag{5}$$

that leads to complicated polynomial relations involving $\kappa$, such as

$$\kappa_{abbm}\kappa_{acdm}\kappa_{dppn}\kappa_{qqcn} + \kappa_{abam}\kappa_{bcdm}\kappa_{dppn}\kappa_{qqcn}$$
$$+2\kappa_{aabm}\kappa_{bcdm}\kappa_{pdpn}\kappa_{qqcn} - 4\kappa_{aabm}\kappa_{bcdm}\kappa_{ppcn}\kappa_{qqdn} = 0\,. \tag{6}$$

The symmetries of $\kappa$ and the Riemann tensor $R$ and often called *multi-term* as they involve equations with more than two terms. One cannot fully account for the peculiar properties of these tensors by simply tracking some sign (or phase) under exchange of indices. Another way of looking at these tensors is to say that, under permutations of their indices, they do not transform as 1-dimension representations of the relevant permutation group. In fact, it is well know that $R$ transforms as the

 $$\tag{7}$$

irreducible representation of $S_4$, which is 2-dimensional. On the other hand, $\kappa$ transforms as the

 $$\tag{8}$$

reducible representation of $S_3 \times S_1$, which has dimension $1 + 2 + 1 = 4$. Even more complicated symmetries arise in effective field theories when one considers operators of higher dimensions, such as those which violate baryon and/or lepton number — see for example [6–9]. Having said this, the reader unfamiliar with the permutation group and its representations shouldn't be overly concerned as the algorithm described in this paper does not rely on it, not does on need to know any of this in order to use the main function of the `SimTeEx` program (to be introduced later), which puts a tensor expression in canonical form. The program also contains some auxiliary functions, described in the appendix A, which do require some knowledge of the permutation group. I have tried to make the discussion there somewhat self-contained, however textbooks on the matter, such as [10], might still come in handy.

The twofold purpose of this work is to (1) describe an algorithm which can take into account arbitrarily complicated symmetry relations, producing a canonical form for a polynomial expression with one or more tensors, and (2) introduce the Mathematica package `SimTeEx` which implements these computations. Although the algorithm discussed in this paper draws no inspiration from previous works, it turns out that it shares similarities with several previous codes:

- Dummy indices are dealt with graphs, which are used to represent tensor monomials in a canonical form, making `SimTeEx` similar to `Redberry` [11] in this aspect.

- Multi-term symmetries are seen as a linear algebra problem to be solved by putting matrices in reduced echelon form, an idea which is akin to the one used in `ATENSOR` [12].

- Columns of these matrices are reordered with the purpose of making sure that the number of monomials of an input expression does not increase in the final result. This is analogous to the `meld` algorithm [13] of `Cadabra` [2].

To this list of packages, I should add `xPerm` [14] (part of `xAct` [15]), which also is capable of simplifying tensor expressions.

Readers only interested in using the `SimTeEx` program may jump directly to section 5. For those interested in the algorithm used by this code,

- section 2 introduces graphs as a way of handling dummy indices, and in doing so they can be used to represent each tensor monomial in a canonical form;

- section 3 briefly discusses the fact that tensor polynomials are a vector space generated by the above mentioned graphs;

- section 4 addresses the difficulty introduced by tensors with permutation symmetries, and equates the task of simplifying expressions with them to putting a matrix in reduced echelon form.

A summary is presented in section 6, together with a discussion of possible future developments for the `SimTeEx` program. In appendix A the reader can find a description of several extra functions which are available in `SimTeEx`, and finally appendix B discusses why Young symmetrizers may not be sufficient to describe the symmetries of a tensor.

## 2. Dummy indices and graphs

Let us ignore for now the possibilities of tensors having permutation symmetries. Graphs are a very suggestive way of representing a product of contracted tensors.[1] For example, considering (wrongly) for a moment that $R$ is a fully symmetric rank-4 tensor, the first term in equation (2) is quite naturally represented by the 5-loop diagram in figure 1.

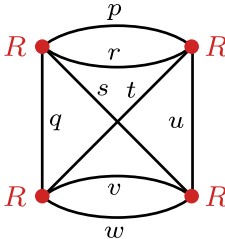

Figure 1: Potential graph representation of $R_{pqrs}R_{ptru}R_{tvqw}R_{uvsw}$, assuming that $R$ is completely symmetric. Edge labels are shown only to make it easier to compare this representation with the original expression.

---

[1] For the more mathematical inclined readers, I should note that one can have pairs of vertices connected with multiple edges (see figures 1 and 2), and/or edges connecting a vertex to itself (as in figure 2). As such, strictly speaking, we are dealing with multigraphs/pseudographs.

Likewise, under the assumption that $\kappa$ is also fully symmetry and ignoring the constant prefactor of 2, the first term in equation (6) has the graph representation show in figure 2.

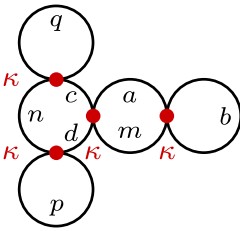

Figure 2: Potential graph representation of $\kappa_{abbm}\kappa_{acdm}\kappa_{dppn}\kappa_{qqcn}$, assuming that $\kappa$ is completely symmetric. As in figure 1, the edges should remain unlabeled (they are only shown here for clarity).

There are several important observations to be made:

1. In these two figures, the edges are tagged with the corresponding dummy indices only to facilitate the comparison with the original tensor expression. In fact, in order to get a representation of tensor monomials which is invariant under relabeling of dummy indices, the graphs edges are to be unlabeled.

2. Not withstanding this last point, I did assume that the participating tensors ($R$ and $\kappa$) were completely symmetric. When this is not the case, it is important to differentiate which indices of the tensors are being contracted, so we do need to label the edges with the slot positions of the corresponding dummy indices. One can do so in practice by assigning a direction to the edge, and registering that information together with the slot positions of the contracting indices (but not their letters/names).

3. The vertices also need to be labeled with the name of the corresponding tensor in order to avoid ambiguities. That's because there might be more than one rank-$n$ tensor in a given monomial.

4. The two example above do not have external indices. When they do exist, an obvious solution is to treat each of them as a vertex in the diagram, with degree/valency 1 (i.e. they connect to the rest of the graph via a single edge).

For instance, $U_{ijk}U_{klm}T_{njlp}$ could be represented by the directed graph shown in figure 3, with labeled vertices and edges.

In this way, the problem of determining if two tensor monomials are the same is translated to the well known problem of determining if two graphs are isomorphic.

Let us finish this discussion on the graphical representation of tensor monomials by briefly observing that obviously, for practical manipulations such as checking for isomorphisms, one must find some way to represent the graphs themselves. The program SimTeEx uses what can be described as a generalized adjacency matrix, which must be significantly more complicated than a normal one in order to track all the information mentioned above. Briefly, if we were to

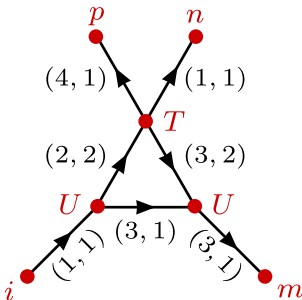

Figure 3: Directed graph representation of $U_{ijk}U_{klm}T_{njlp}$. Edge labels $(n_1, n_2)$ indicate that the index in slot $\#n_1$ of the departing tensor contracts with the index in slot $\#n_2$ of the incident tensor. For example the $(3,2)$ implies that the third index of $T$ contract with the second index of one of the $U$ tensors (the one with the external $m$ index). It is important to give a direction to the edges so that one can interpret the two numbers which label each edge.

remove all the labels and arrows in figure 3's diagram, the resulting graph, with degree sequence (4,3,3,1,1,1,1), could be represented by the matrix

$$
\mathscr{M} = \begin{pmatrix}
0 & 1 & 1 & 1 & 1 & 0 & 0 \\
1 & 0 & 1 & 0 & 0 & 1 & 0 \\
1 & 1 & 0 & 0 & 0 & 0 & 1 \\
1 & 0 & 0 & 0 & 0 & 0 & 0 \\
1 & 0 & 0 & 0 & 0 & 0 & 0 \\
0 & 1 & 0 & 0 & 0 & 0 & 0 \\
0 & 0 & 1 & 0 & 0 & 0 & 0
\end{pmatrix}
\tag{9}
$$

whose entry $(i, j)$ indicates how many connections are there between vertex $\#i$ and vertex $\#j$. For an undirected graph, this matrix is symmetric, while for directed ones the entry $(i, j)$ indicates the number of edges going from vertex $\#i$ to vertex $\#j$, which might differ from the value of entry $(j, i)$. To represent labeled vertices, instead of representing the graph just with $\mathscr{M}$, one can add a list with the vertex labels (matching the ordering in the adjacency matrix): graph $= \{\{T, U, U, p, n, i, m\}, \mathscr{M}\}$. As for the edge labels which identify the positions of tensor entries under contraction, one possibility is the following: if vertex $x$ corresponds to an $r_x$-rank tensor, then one can promote entry $(\alpha, \beta)$ of $\mathscr{M}$ to an $r_\alpha \times r_\beta$ dimensional matrix, such that its entry $(a, b)$ is equal to 1 if index $\#a$ of the tensor associated to vertex $\alpha$ is contracted with index $\#b$ of the tensor associated to vertex $\beta$, and 0 otherwise. For example, the monomial $U_{ijk}U_{klm}T_{njlp}$

is fully described by $\left\{ \{T, U, U, p, n, i, m\}, \mathscr{M}^{(\text{gen})} \right\}$, with

$$
\mathscr{M}^{(\text{gen})} = \left(
\begin{array}{ccccccc}
\begin{pmatrix} 0\,0\,0\,0 \\ 0\,0\,0\,0 \\ 0\,0\,0\,0 \\ 0\,0\,0\,0 \end{pmatrix} &
\begin{pmatrix} 0\,0\,0 \\ 0\,1\,0 \\ 0\,0\,0 \\ 0\,0\,0 \end{pmatrix} &
\begin{pmatrix} 0\,0\,0 \\ 0\,0\,0 \\ 0\,1\,0 \\ 0\,0\,0 \end{pmatrix} &
\begin{pmatrix} 0 \\ 0 \\ 0 \\ 1 \end{pmatrix} &
\begin{pmatrix} 1 \\ 0 \\ 0 \\ 0 \end{pmatrix} &
\begin{pmatrix} 0 \\ 0 \\ 0 \\ 0 \end{pmatrix} &
\begin{pmatrix} 0 \\ 0 \\ 0 \\ 0 \end{pmatrix} \\[1.5em]
\begin{pmatrix} 0\,0\,0\,0 \\ 0\,1\,0\,0 \\ 0\,0\,0\,0 \end{pmatrix} &
\begin{pmatrix} 0\,0\,0 \\ 0\,0\,0 \\ 0\,0\,0 \end{pmatrix} &
\begin{pmatrix} 0\,0\,0 \\ 0\,0\,0 \\ 1\,0\,0 \end{pmatrix} &
\begin{pmatrix} 0 \\ 0 \\ 0 \end{pmatrix} &
\begin{pmatrix} 0 \\ 0 \\ 0 \end{pmatrix} &
\begin{pmatrix} 1 \\ 0 \\ 0 \end{pmatrix} &
\begin{pmatrix} 0 \\ 0 \\ 0 \end{pmatrix} \\[1.5em]
\begin{pmatrix} 0\,0\,0\,0 \\ 0\,0\,1\,0 \\ 0\,0\,0\,0 \end{pmatrix} &
\begin{pmatrix} 0\,0\,1 \\ 0\,0\,0 \\ 0\,0\,0 \end{pmatrix} &
\begin{pmatrix} 0\,0\,0 \\ 0\,0\,0 \\ 0\,0\,0 \end{pmatrix} &
\begin{pmatrix} 0 \\ 0 \\ 0 \end{pmatrix} &
\begin{pmatrix} 0 \\ 0 \\ 0 \end{pmatrix} &
\begin{pmatrix} 0 \\ 0 \\ 0 \end{pmatrix} &
\begin{pmatrix} 0 \\ 0 \\ 1 \end{pmatrix} \\[1.5em]
\begin{pmatrix} 0\,0\,0\,1 \end{pmatrix} &
\begin{pmatrix} 0\,0\,0 \end{pmatrix} &
\begin{pmatrix} 0\,0\,0 \end{pmatrix} &
\begin{pmatrix} 0 \end{pmatrix} &
\begin{pmatrix} 0 \end{pmatrix} &
\begin{pmatrix} 0 \end{pmatrix} &
\begin{pmatrix} 0 \end{pmatrix} \\[1em]
\begin{pmatrix} 1\,0\,0\,0 \end{pmatrix} &
\begin{pmatrix} 0\,0\,0 \end{pmatrix} &
\begin{pmatrix} 0\,0\,0 \end{pmatrix} &
\begin{pmatrix} 0 \end{pmatrix} &
\begin{pmatrix} 0 \end{pmatrix} &
\begin{pmatrix} 0 \end{pmatrix} &
\begin{pmatrix} 0 \end{pmatrix} \\[1em]
\begin{pmatrix} 0\,0\,0\,0 \end{pmatrix} &
\begin{pmatrix} 1\,0\,0 \end{pmatrix} &
\begin{pmatrix} 0\,0\,0 \end{pmatrix} &
\begin{pmatrix} 0 \end{pmatrix} &
\begin{pmatrix} 0 \end{pmatrix} &
\begin{pmatrix} 0 \end{pmatrix} &
\begin{pmatrix} 0 \end{pmatrix} \\[1em]
\begin{pmatrix} 0\,0\,0\,0 \end{pmatrix} &
\begin{pmatrix} 0\,0\,0 \end{pmatrix} &
\begin{pmatrix} 0\,0\,1 \end{pmatrix} &
\begin{pmatrix} 0 \end{pmatrix} &
\begin{pmatrix} 0 \end{pmatrix} &
\begin{pmatrix} 0 \end{pmatrix} &
\begin{pmatrix} 0 \end{pmatrix}
\end{array}
\right). \tag{10}
$$

Taking a look at the block $(\alpha, \beta) = (1, 3)$ of this large matrix,

$$
\mathscr{M}_{1,3}^{(\text{gen})} = \begin{pmatrix} 0\,0\,0 \\ 0\,0\,0 \\ 0\,1\,0 \\ 0\,0\,0 \end{pmatrix}, \tag{11}
$$

one can see how vertex $\alpha = 1$ (a $T$ tensor) contracts with vertex $\beta = 3$ (a $U$ tensor): the '1' in the $(a, b) = (3, 2)$ entry implies that the 3rd index of the $T$ tensor contracts to the 2nd index of the $U$ tensor.

Since there are only two vertices with the same tensor name, the 2nd and the 3rd in the list above, the equivalence of $U_{ijk}U_{klm}T_{njlp}$ with some other $U_{...}U_{...}T_{....}$ monomial can be checked with $2! = 2$ row and column permutations of the generalized adjacency matrix. Rather than do this for every pair of graphs $g = \left\{ \langle \text{tensor list } \mathscr{T} \rangle, \langle \text{generalized adjacency matrix } \mathscr{M}^{(\text{gen})} \rangle \right\}$ and $g' = \left\{ \langle \text{tensor list } \mathscr{T} \rangle, \langle \text{generalized adjacency matrix } \mathscr{M}'^{(\text{gen})} \rangle \right\}$ to be compared, it is more practical to put each graph $g$ in a canonical form $g_{\text{canonical}}$ by performing all row/column permutations leading to equivalent graphs and picking out from this list a representative.[2] Two monomials are then equal if and only if their representative is the same.

---

[2]For example, one can do this by first sorting the list of tensors $\mathscr{T}$ (and reordering accordingly the rows and columns of $\mathscr{M}^{(\text{gen})}$ while doing so). Then, from all the equivalent $g_{\pi_i} = \left\{ \langle \text{sorted tensor list } \mathscr{T} \rangle, \mathscr{M}_{\pi_i}^{(\text{gen})} \right\}$ one can take $g_{\text{canonical}}$ to be the one associated to the smallest $\mathscr{M}_{\pi_i}^{(\text{gen})}$, under some sorting criteria.

# 3. Polynomials as a vector space

Quite naturally, one can see a polynomial $\boldsymbol{P}$ involving tensors as being a linear combination of the graphs $g_i$ described above, with numerical or symbolic coefficients $c_i$. The distinguishing feature of these $c_i$ is simply that they do not involve tensors (i.e. objects with indices), for example the $7x$ in $7xU_{ijk}U_{klm}T_{njlp}$. Thus, a tensor polynomial $\boldsymbol{P}$ can be seen as a vector

$$\boldsymbol{P} = \sum_i c_i g_i \tag{12}$$

in a vector-space spanned by several $g_i$, and the discussion above provides a fail-proof method of checking if any of two elements of this generating set are equivalent. Indeed, if some $g_1$ is found to be equivalent to some $g_2$, we can drop one of these vectors from the generating set, and add the corresponding coefficients:

$$c_1 g_1 + c_2 g_2 + \cdots \rightarrow (c_1 + c_2)\, g_1 + \cdots . \tag{13}$$

As an example, consider the tensor polynomial `2 m[a,q,q,b] T[a,b] + A m[c1,m,m,c2] T[c1,c2]` which one can represent as $2g_1 + Ag_2$. Given the simplicity of this example, and the absence of symmetries, it is clear already by eye that the two $g$'s are equivalent:

$$g_1 = g_2 = \left\{ \{m, T\}, \left( \begin{pmatrix} 0 & 0 & 0 & 0 \\ 0 & 0 & 1 & 0 \\ 0 & 1 & 0 & 0 \\ 0 & 0 & 0 & 0 \end{pmatrix} \begin{pmatrix} 1 & 0 \\ 0 & 0 \\ 0 & 0 \\ 0 & 1 \end{pmatrix} \\ \begin{pmatrix} 1 & 0 & 0 & 0 \\ 0 & 0 & 0 & 1 \end{pmatrix} \begin{pmatrix} 0 & 0 \\ 0 & 0 \end{pmatrix} \right) \right\} . \tag{14}$$

This means that the original expression can be simplified to a single term: perhaps `(2+A) m[a,q,q,b] T[a,b]` or `(2+A) m[c1,m,m,c2] T[c1,c2]`. These two possibilities are quite natural, as they reuse the index labels provided in the input, however one should keep in mind that the choice of dummy indices in the final result is arbitrary. Indeed, one could discard altogether the ones in the input expression, insisting on putting the dummy indices in some canonical form — such as $i_1, i_2, \cdots$ — yielding expressions of the type `(2+A) m[i1,i2,i2,i3] T[i1,i3]`.

# 4. Tensors with symmetries

The discussion so far concerns exclusively the problem introduced by dummy indices and how to address it by using graphs to represent tensor monomials. Things becomes more complicated when there are symmetries under exchange of indices.

Before diving into these complications, one ought to distinguish the 'simple' symmetries from the 'hard' ones. The simple ones, often called *monoterm symmetries*, are those which can be accounted for by tracking just a complex phase $\sigma$ (which often is just a $\pm$ sign):

$$T_{\pi(i_1 i_2 \cdots i_n)} = \sigma T_{i_1 i_2 \cdots i_n} . \tag{15}$$

Here $\pi$ stands for some permutation of the indices, and $\sigma$ must be an $m$th root of unity ($\sigma^m = 1$) with $m$ being the order of the permutation $\pi$, i.e. $\pi^m = e$ (the identity). Here are four examples:

$$T_{ab} = -T_{ba}\,, \tag{16}$$

$$T_{abcd} = T_{badc}\,, \tag{17}$$

$$T_{abcd} = T_{bacd} \text{ and } T_{abcd} = -T_{abdc}\,, \tag{18}$$

$$T_{abc} = \omega T_{bca} \text{ with } \omega \equiv \exp\left(\frac{2\pi i}{3}\right)\,. \tag{19}$$

In each of these cases, $T$ is no longer a general tensor, as it obeys some symmetry under which permuting the indices gives back the original tensor, up to some phase factor. It is very significant that the group describing these permutations is abelian in all four cases: $Z_2$, $Z_2$, $Z_2 \times Z_2$ and $Z_3$, respectively. That's because the irreducible representations of abelian groups are 1-dimensional, so their action on the indices of some tensor can always be reduced to studying a phase (as in equation (15)).

Incorporating tensors with monoterm symmetries in the graph formalism is straightforward: two monomials are proportional to each other if they are represented by equivalent graphs, where equivalence is established not only by permuting equal vertices but also by permuting the edges associated to the monoterm symmetries of each tensor (i.e. the vertices of the graphs). Note that it is necessary to track a phase $\sigma$ every time these edge permutations are performed.

Let us then turn our attention to the 'hard' symmetries — the so-called *multiterm symmetries* — which are of the form

$$T_{\pi_1(i_1 i_2 \cdots i_n)} + T_{\pi_2(i_1 i_2 \cdots i_n)} + \cdots T_{\pi_p(i_1 i_2 \cdots i_n)} = 0 \text{ with } p > 2\,. \tag{20}$$

For concreteness, consider a tensor $T$ with the symmetry

$$T_{abc} + T_{bca} + T_{cab} = 0 \tag{21}$$

and, putting aside the problem of dummy indices, the very basic expression

$$x T_{abc} + y T_{bca} + z T_{cab} \tag{22}$$

which we wish to simplify. One may establish some ordering of least to most desired form of a tensor, for example $T_{abc} < T_{bca} < T_{cab}$, in which case we would prioritize eliminating all instances of $T_{abc}$:

$$x T_{abc} + y T_{bca} + z T_{cab} \to (y - x)\, T_{bca} + (z - x)\, T_{cab}\,. \tag{23}$$

From this very basic example, we are already in a position to make some key remarks:

1. Clearly, the simplified expression depends on the (arbitrary) order of preference among the various permutations of $T$.

2. By having any fixed order of preference among the various permutations of $T$, we may be forced to have a result with more monomials than we started with. That is what happens in expression (23) for $x = 1$ and $y = z = 0$. In other words, in the presence of multiterm symmetries, putting an expression in a *canonical form* may not necessarily mean the same as *simplifying* it (see also [13]).

3. In the expression $U_{abc}\left(xT_{abc} + yT_{bca} + zT_{cab}\right)$, where a second tensor $U$ with no symmetries was introduced, all indices are summed over and as such they can be freely relabeled. Therefore it is meaningless to even consider an order between $T_{abc}$, $T_{bca}$ and $T_{cab}$. Nonetheless, one can still order the monomials $U_{abc}T_{abc}$, $U_{abc}T_{bca}$ and $U_{abc}T_{bca}$, by using a graph representation (as detailed in section 2).

With these insights, one may reduce the task of simplifying expressions with multiterm symmetries to a linear algebra problem. The input is some tensor polynomial $\boldsymbol{P} = \sum_{i=1}^{k} c_i g_i$ where each of the $k$ terms is represented by a coefficient $c_i$ and a graph $g_i$, which we can assume from now on to be in a canonical form. Multiterm symmetries are relations of the form

$$0 = \sum_{i=1}^{k} n_i^{(a)} g_i \tag{24}$$

where the index $a$ accounts for the possible existence of several relations. Some of the $c_i$ coefficients of the original expression $\boldsymbol{P}$ can be converted to zero by adding to $\boldsymbol{P}$ multiples $\omega_a$ of $\sum_{i=1}^{k} n_i^{(a)} g_i$:

$$\boldsymbol{P} = \sum_{i=1}^{k} c_i g_i \rightarrow \boldsymbol{P}' = \boldsymbol{P} + \sum_{a} \omega_a \sum_{i=1}^{k} n_i^{(a)} g_i \equiv \sum_{i=1}^{k} c_i' g_i \,. \tag{25}$$

Many readers will probably consider $\boldsymbol{P}'$ to be in the simplest form when there are as many zero $c_i'$ coefficient as possible. A rather straightforward and efficient way of nullifying some $c_i'$ coefficients is to put the matrix

$$
\overbrace{\hspace{2.2cm}}^{\begin{matrix} g_1 & g_2 & \cdots & g_k \end{matrix}}
\begin{pmatrix}
1 & c_1 & c_2 & \cdots & c_k \\
\hline
0 & n_1^{(1)} & n_2^{(1)} & \cdots & n_k^{(1)} \\
0 & n_1^{(2)} & n_2^{(2)} & \cdots & n_k^{(2)} \\
\vdots & \vdots & \vdots & & \ddots
\end{pmatrix}
\tag{26}
$$

in reduced row echelon form (RREF) and then take $c_1'$, $c_2'$, ...,$c_k'$ from columns 2 to $k+1$ of the first line.[3] Note that the number of null coefficients will depend on the ordering of the columns of this matrix (i.e. the ordering of the $g_i$). In the case of expression (22) and a $T$ tensor with the symmetry (21) we would get from this procedures the coefficients $(c_1', c_2', c_3') = (0, y - x, z - x)$ for each monomial, assuming an ordering $T_{abc} < T_{bca} < T_{cab}$:

$$
\overbrace{\begin{pmatrix} 1 & x & y & z \\ \hline 0 & 1 & 1 & 1 \end{pmatrix}}^{\begin{matrix} T_{abc} & T_{bca} & T_{cab} \end{matrix}} \xrightarrow{\text{RREF}} \overbrace{\begin{pmatrix} 1 & 0 & y - x & z - x \\ \hline 0 & \cdots & \cdots & \cdots \end{pmatrix}}^{\begin{matrix} T_{abc} & T_{bca} & T_{cab} \end{matrix}} \,.
\tag{27}
$$

More symmetries — of the same tensor or perhaps others — can be taken into account by simply adding extra lines to this matrix. Note that the order of the graphs (i.e. the matrix columns) is important, and to maximize the number of null coefficients one would have to test all possible column orderings. Since the number of relevant terms can be quite large, in general it would not be feasible to test all column ordering. In our very simple example, using the same ordering as in

---

[3]The first column in (26) is introduced only to make sure that the first line remains at the top.

(27), for $x = 0$ and $y = z = 1$ we would get no change in the final result, i.e. $(c_1', c_2', c_3') = (0, 1, 1)$. However, swapping the positions of $T_{abc}$ and $T_{bca}$, we would get a single term:

$$\overbrace{\begin{pmatrix} 1 & | & 1 & 0 & 1 \\ \hline 0 & | & 1 & 1 & 1 \end{pmatrix}}^{T_{bca}\ T_{abc}\ T_{cab}} \xrightarrow{\text{RREF}} \overbrace{\begin{pmatrix} 1 & | & 0 & -1 & 0 \\ \hline 0 & | & \cdots & \cdots & \cdots \end{pmatrix}}^{T_{bca}\quad T_{abc}\quad T_{cab}}. \tag{28}$$

Setting aside the ambitious goal of always having a minimum number of monomials, as already mentioned above, there is the related issue of potentially having more terms in the final result than in the input expression. However, it is rather simple to avoid this outcome. All that is needed is for the columns associated to terms which do exist in the input expression (i.e. those for which $c_i \neq 0$) to appear last. In the example $(c_1, c_2, c_3) = (1, 0, 0)$, which leads to $(c_1', c_2', c_3') = (0, -1, -1)$,

$$\overbrace{\begin{pmatrix} 1 & | & 1 & 0 & 0 \\ \hline 0 & | & 1 & 1 & 1 \end{pmatrix}}^{T_{abc}\ T_{bca}\ T_{cab}} \xrightarrow{\text{RREF}} \overbrace{\begin{pmatrix} 1 & | & 0 & -1 & -1 \\ \hline 0 & | & \cdots & \cdots & \cdots \end{pmatrix}}^{T_{abc}\quad T_{bca}\quad T_{cab}}, \tag{29}$$

one would swap the first and third columns; from $(c_1, c_2, c_3) = (0, 0, 1)$ we would get an unchanged result $(c_1', c_2', c_3') = (0, 0, 1)$:

$$\overbrace{\begin{pmatrix} 1 & | & 0 & 0 & 1 \\ \hline 0 & | & 1 & 1 & 1 \end{pmatrix}}^{T_{cab}\ T_{bca}\ T_{abc}} \xrightarrow{\text{RREF}} \overbrace{\begin{pmatrix} 1 & | & 0 & 0 & 1 \\ \hline 0 & | & \cdots & \cdots & \cdots \end{pmatrix}}^{T_{cab}\quad T_{bca}\quad T_{abc}}. \tag{30}$$

In more elaborate cases, the $c_i'$ would not necessarily be the same as the input $c_i$. Crucially, it is impossible for the number of non-zero coefficients to grow. The end effect of this reshuffling of columns is similar to the one achieved by the `meld` algorithm of `Cadabra 2` [13].

# 5. Using `SimTeEx`

## The main function

Readers which are mainly interested in using the `SimTeEx` program may do so by first downloading it from

renatofonseca.net/simteex

then installing and loading the program in Mathematica,[4]

`<<SimTeEx`` `

and finally calling the function

---

[4]In order to run the extra functions mentioned in appendix A it is also necessary to have `GroupMath` [16] installed in the system.

```
CanonicalForm[<tensor polynomial>,<symmetries>]
```

Simplifies the given polynomial expression. The second argument (which is optional) should
be a list of null expressions encoding the symmetries of the various tensors.

For example:

```
In[•]:= CanonicalForm[T[i, a] ×U[a, j] + T[i, b] ×U[b, j]]
        CanonicalForm[α T[i, a] ×U[a, j] + β T[i, a] ×U[j, a], {U[x, y] +U[y, x]}]

Out[•]= 2 T[i, a] ×U[a, j]

Out[•]= (α − β) T[i, a] ×U[a, j]
```

The last argument in the second example tells the program that the $U$ tensor has the symmetry
$U_{xy} + U_{yx} = 0$, i.e. it is anti-symmetric. The user is free to choose the names of tensors and indices,
both in the expression to simplify as well as in the symmetry conditions. Everything with a head
and square brackets, `head[...]`, is assumed to be a tensor, and the variables inside the brackets
are taken to be indices. As such, there is no need to declare the list of tensors and indices to be
used.

Equation (2) involving the Riemann tensor can be checked with the following code:

```
In[•]:= expressionToSimplify = R[p, q, r, s] ×R[p, t, r, u] ×R[t, v, q, w] ×R[u, v, s, w] −
          R[p, q, r, s] ×R[p, q, t, u] ×R[r, v, t, w] ×R[s, v, u, w] −
          R[m, n, a, b] ×R[n, p, b, c] ×R[m, s, c, d] ×R[s, p, d, a] +
          x R[m, n, a, b] ×R[p, s, b, a] ×R[m, p, c, d] ×R[n, s, d, c];
        symmetries = {R[f1, f2, f3, f4] +R[f2, f1, f3, f4], R[f1, f2, f3, f4] +R[f1, f2, f4, f3],
          R[f1, f2, f3, f4] +R[f1, f3, f4, f2] +R[f1, f4, f2, f3]};
        CanonicalForm[expressionToSimplify, symmetries]

Out[•]= 1/8 (−2 + 8 x) R[m, n, a, b] ×R[m, p, c, d] ×R[n, s, d, c] ×R[p, s, b, a]
```

The factor $1/4$ in (2) was intentionally replaced with a generic $x$ to make clear, from the output,
that the expression is null only for $x = 1/4$.

The user does not need to know what is the representation of the permutation group under
which the tensors transform. In fact the user is free to provide a list of symmetries which makes
little sense, such as

```
In[•]:= CanonicalForm[x1 H[i, j] ×T[j, k] + x2 H[i, a] ×T[k, a], {T[a, b] − 2 T[b, a]}]

Out[•]= 0
```

The zero is explained by the fact that a tensor with the symmetry $T_{ab} - 2T_{ba} = 0$ is necessarily
null ($T_{ab} = 0$). The procedure described in section 4 is very flexible, handling well these situations
without any need for special code.

`Cadabra` [2] — the only other code supporting multi-term symmetries which I am aware of
— requires that the user indicate a Young symmetrizer for each tensor. Asking only for a set of

symmetry equations (like those in equation (1)) may offer some advantages. Firstly, while this information is readily available for well known tensors, in general the user must figure out the Young tableau associated to a particular set of symmetry relations. On this topic, I will note that the user is free to provide as input to `SimTeEx` any equivalent set of equations (see also the function `SameEquationsQ` in appendix A). For example, in the case above involving the Riemann tensor, one could rewrite it as follows:

```
        symmetries2 = {R[f1, f2, f3, f4] + R[f3, f4, f2, f1],
            R[f1, f2, f3, f4] + R[f1, f3, f4, f2] + R[f1, f4, f2, f3]};
        CanonicalForm[expressionToSimplify, symmetries2]
```
$$Out[\circ]= \frac{1}{8} \; (-2 + 8\,x) \; R[m, n, a, b] \times R[m, p, c, d] \times R[n, s, d, c] \times R[p, s, b, a]$$

Secondly, the symmetries of some tensors, such as $\kappa_{ijkl}$ mentioned in the introduction of this work, involve a reducible representation of the relevant permutation group and therefore they are described by the sum of several Young symmetrizers. Finally, one can have tensors with symmetries that cannot be described with Young symmetrizers at all (readers interested in this topic are referred to appendix B for details).

One should be aware that the function `CanonicalForm` when applied to two equivalent expressions, $\text{expr}_1$ and $\text{expr}_2$, may yield different outputs,

$$\texttt{CanonicalForm}\,(\text{expr}_1) \neq \texttt{CanonicalForm}\,(\text{expr}_2)\,, \tag{31}$$

although it is always be true that

$$\texttt{CanonicalForm}\,(\text{expr}_1 - \text{expr}_2) = 0\,. \tag{32}$$

The reason for the inequality of outputs (31) is twofold:

1. To ensure that the number of terms never increases, the program performs the sorting operation which is mentioned at the end of section 4. Since this operation depends on the terms which do appear in the input one might have $\texttt{CanonicalForm}\,(\text{expr}_1) \neq \texttt{CanonicalForm}\,(\text{expr}_2)$ even if the two expressions are the equivalent.

2. An even simpler reason is that `CanonicalForm` reuses the labels for dummy indices given in the input, in order not to introduce new ones which are unfamiliar to the user. Therefore in the trivial example where $\text{expr}_1 = T_i T_i$ and $\text{expr}_2 = T_j T_j$ the function does not change the inputs at all, i.e. $\texttt{CanonicalForm}\,(\text{expr}_1) = \text{expr}_1 = T_i T_i$ which is clearly different from $\texttt{CanonicalForm}\,(\text{expr}_2) = \text{expr}_2 = T_j T_j$. On the other hand, $\texttt{CanonicalForm}\,(T_i T_i - T_j T_j) = 0$.

For some authors [13, 17] a function with these properties, (31) and (32), calculates the *normal form* rather than the *canonical form* of an expression. The user can change the default behavior of `CanonicalForm` described above by setting the global flag `$TrueCanonicalForm` to `True` (the default value is `False`):

```
In[ ]:= $TrueCanonicalForm = True;
       sym = {T[a, b, c] + T[b, c, a] + T[c, a, b]};

       CanonicalForm[T[c, a, b] × X[a, b], sym]
       CanonicalForm[(-T[a, b, c] - T[b, c, a]) X[a, b], sym]
       % === %%

Out[ ]= -T[f1, f2, c] × X[f1, f2] - T[f1, c, f2] × X[f2, f1]

Out[ ]= -T[f1, f2, c] × X[f1, f2] - T[f1, c, f2] × X[f2, f1]

Out[ ]= True
```

With $TrueCanonicalForm=True the dummy indices are picked from a list in another global flag, $CanonicalListOfIndices, which the user can change at will (the default value is {f1,f2,f3,f4, f5,f6,...}).

## Expressions with anti-commuting tensors

In high energy physics one often needs to handle fermion fields which behave as Grassmann numbers. Being non-trivial representations of the Lorentz group, they have at the very least a spinor index; very often they have others, such as gauge and flavor indices. For this reason, it is useful to consider polynomials where some tensors anti-commute. SimTeEx is prepared to handle such cases: on one hand, all fermionic tensors must be declared in the global variable $CanonicalFormFermions; on the other hand, since the order of the factors is now important, the built in command NonCommutativeMultiply (**) must be used in the expression to simplify.

Consider for example a mass term

$$m_{ij}\epsilon_{\alpha\beta}\psi_i^\alpha\psi_j^\beta \tag{33}$$

for Weyl spinors $\psi$, where the $\alpha, \beta = 1, 2$ are spinor indices which contract with a Levi-Civita tensor $\epsilon$ to form a Lorentz invariant, while $i, j$ are flavor indices. It is well known that the mass matrix $m$ is symmetric (or, more rigorously, that only its symmetric part contributes to the above expression); this is only true because the spinors $\psi$ anticommute. One can check with SimTeEx that $(m_{ij} - m_{ji})\,\epsilon_{\alpha\beta}\psi_{\alpha,i}\psi_{\beta,j} = 0$ as follows:

```
In[ ]:= $CanonicalFormFermions = {ψ};

       expression1 = ϵ[α, β] × ψ[α, i] ** ψ[β, j] × m[i, j];
       expression2 = ϵ[α, β] × ψ[α, i] ** ψ[β, j] × m[j, i];
       CanonicalForm[expression1 - expression2, {ϵ[f1, f2] + ϵ[f2, f1]}]

Out[ ]= 0
```

As a more complicated example, consider the dimension 6 operator obtained from squaring the one above, i.e.

$$\mathcal{O}_{ijkl} \equiv \left(\psi_i^T \epsilon \psi_j\right)\left(\psi_k^T \epsilon \psi_l\right) = \epsilon_{\alpha\beta}\epsilon_{\gamma\delta}\psi_i^\alpha\psi_j^\beta\psi_k^\gamma\psi_l^\delta \,. \tag{34}$$

Calling $\epsilon^{\mathrm{sq}}_{\alpha\beta\gamma\delta}$ to $\epsilon_{\alpha\beta}\epsilon_{\gamma\delta}$, it is important to note that on top of the obvious symmetries $\epsilon^{\mathrm{sq}}_{\alpha\beta\gamma\delta} = -\epsilon^{\mathrm{sq}}_{\beta\alpha\gamma\delta} = \epsilon^{\mathrm{sq}}_{\gamma\delta\alpha\beta}$, this tensor also obeys the equation $\epsilon^{\mathrm{sq}}_{\alpha\beta\gamma\delta} + \epsilon^{\mathrm{sq}}_{\alpha\delta\beta\gamma} + \epsilon^{\mathrm{sq}}_{\alpha\gamma\delta\beta} = 0$.[5] Using these properties we can derive that

$$\mathcal{O}_{ijkl} = \mathcal{O}_{jikl} = \mathcal{O}_{klij} \text{ and } \mathcal{O}_{ijkl} + \mathcal{O}_{iklj} + \mathcal{O}_{iljk} = 0\,. \tag{35}$$

```
In[ ]:= symmetriesOfεε = {εε[α, β, γ, δ] + εε[β, α, γ, δ], εε[α, β, γ, δ] - εε[γ, δ, α, β],
          εε[α, β, γ, δ] + εε[α, δ, β, γ] + εε[α, γ, δ, β]};
       Oijkl = ψ[α, i] ** ψ[β, j] ** ψ[γ, k] ** ψ[δ, l]  εε[α, β, γ, δ];
       Ojikl = Oijkl /. {i → j, j → i};
       Oklij = Oijkl /. {i → k, j → l, k → i, l → j};
       Oiklj = Oijkl /. {j → k, k → l, l → j};
       Oiljk = Oijkl /. {j → l, k → j, l → k};
       CanonicalForm[Oijkl - Ojikl, symmetriesOfεε]
       CanonicalForm[Oijkl - Oklij, symmetriesOfεε]
       CanonicalForm[Oijkl + Oiklj + Oiljk, symmetriesOfεε]

Out[ ]= 0

Out[ ]= 0

Out[ ]= 0
```

## Alternative format for tensor symmetries

Providing the tensor symmetries as equations is the most general input format accepted by `SimTeEx`. However, if so desired, it is possible to tell the program that some set of indices of a tensor are fully symmetric or antisymmetric with the alternative format `{<tensor head>,<list of indices>, <1 (for sym) or -1 (for antisym)>}`:

```
In[ ]:= CanonicalForm[(x1 S[a, b, c] + x2 S[b, a, c] + x3 S[b, c, a]) T[a, b, c],
          {{S, {1, 2, 3}, 1}}]
       CanonicalForm[(x1 A[a, b, c, d] + x2 A[b, a, c, d] + x3 A[b, c, a, d] + x4 A[a, b, d, c])
          T[a, b, c], {{A, {1, 2, 3}, -1}}]

Out[ ]= (x1 + x2 + x3) S[a, b, c] × T[a, b, c]

Out[ ]= (x1 - x2 + x3) A[a, b, c, d] × T[a, b, c] + x4 A[a, b, d, c] × T[a, b, c]
```

In the case of a tensor which is not necessarily fully symmetric or antisymmetric but nonetheless has monoterm symmetries, of the form in equation (15), this can be indicated with the format `{<tensor head>,Cycles[...], <phase sigma>}`:

---

[5]For numerical tensors (such as $\epsilon^{\mathrm{sq}}_{\alpha\beta\gamma\delta}$) one can find systematically all its symmetries by explicitly comparing the tensor with all its permuted forms; see the function `SymmetriesOfNumericalTensor` in subsection (A.5).

```
In[ ]:= CanonicalForm[x1 T[a, b, c, d] + x2 T[c, d, a, b], {{T, Cycles[{{1, 3}, {2, 4}}], -1}}]
        CanonicalForm[x1 T[a, b, c, d] + x2 T[b, c, d, a], {{T, Cycles[{{1, 2, 3, 4}}], 𝕚}}]

Out[ ]= (x1 - x2) T[a, b, c, d]

Out[ ]= (x1 + 𝕚 x2) T[a, b, c, d]
```

Internally the code converts these alternative input formats into a list of symmetry equations.

### Manually listing the tensor names

For the user's convenience, the program automatically identifies all tensors in the given expression by looking for square brackets. This may sometimes cause problems: for example one cannot use a coefficient named `x[1]` (it would be recognized as a tensor, with a non-symbolic index, leading to an error). On the other hand, some quantities with no visible square brackets are internally represented with them; that is what happens to $\sqrt{2}$, which stands for `Sqrt[2]`. While the particular case of square roots was explicitly addressed in the code, there might be other objects which create difficulties to the user. In order to mitigate this issue, the user can manually provide a list of tensor heads as shown in the following example:

```
In[ ]:= CanonicalForm[x[1] H[i, j] × T[j, k] + x[2] H[i, a] × T[k, a], {T[f1, f2] - T[f2, f1]},
        ListOfHeads → {T, H}]

Out[ ]= H[i, j] × T[j, k] (x[1] + x[2])
```

## 6. Summary

On often encounters polynomial expressions with tensors — some of which have symmetries — in several areas of research, such as in general relativity and in particle physics. It is therefore useful to have a tool that reliably and automatically simplifies these expressions. In this work I have presented an algorithm to do so for arbitrarily complicated tensor symmetries, including the so-called multiterm ones. It is implemented in the `CanonicalForm` function of the `SimTeEx` Mathematica package, which was designed to be as simple as possible to use. The current version of the code also contains five extra functions, described in appendix A, to analyze, compare and transform the symmetries of tensors.

The computational performance of the algorithm and the code described in this work is an important aspect to take into account. Since the equivalence of two tensor monomials can be equated to finding out if two graphs are isomorphic, the problem addressed in this paper is potentially NP-hard. As such, one should not expect to be able to simplify tensor expressions with too many indices. Nevertheless, and while speed was not a major design consideration, some simple modifications to algorithm presented in this work were already implemented in `SimTeEx` with the goal of improving the computational time.

# Acknowledgments

This work was originally developed as a means to expedite calculations for another research project [18], and I would like to thank José Santiago for testing over and over what eventually became the `SimTeEx` code. I am equally grateful to Ricardo Cepedello and Javi F. Martin for reading draft versions of this text, and in Ricardo's case also for testing the code. Additionally, I would like to acknowledge interesting discussions I've had with John Gargalionis and Anders Eller Thomsen on simplifying tensor expressions, plus thank Zhe Ren and Chang-Yuan Yao for pointing out a mistake in an earlier version of this manuscript.

I acknowledge the financial support from the *Consejería de Universidad, Investigación e Innovación*, the Spanish government and the European Union – NextGenerationEU through grant number AST22_6.5; from MCIN/AEI (10.13039/501100011033) through grants number PID2019-106087GB-C22 and PID2022-139466NB-C21; and from the Junta de Andalucía through grant number P21_00199 (FEDER).

# A. Extra tools

Besides the main function — `CanonicalForm` — the `SimTeEx` package contains extra code to analyze and process tensors with symmetries. To ensure that it works properly, the `GroupMath` [16] package needs to be installed in the user's computer, in an appropriate folder, such that it can be loaded automatically by `SimTeEx` with the command $<<$`GroupMath`.

The usage of these extra tools may require some knowledge of the permutation group $S_n$ and its representations. What is mentioned in [16, 19] is sufficient, but in any case, as they become necessary, I lay out below the most salient group theory aspects to have in mind.

For starters, the reader should recall that the permutation group of $n$ objects ($S_n$) contains $n!$ permutations $\pi$. Each of them can be expressed in the cycle notation: for example $\pi = (132)$ is the permutation which replaces object 1 with object 3, object 3 with object 2 and object 2 with object 1: $\{1, 2, 3\} \to \{3, 1, 2\}$. An algebra is formed when we consider linear combinations $\sum_{\pi \in S_n} c_\pi \pi$, as we can multiply ($\star$) two elements of this space using the group multiplication ($\cdot$), namely $\left(\sum_{\pi \in S_n} a_\pi \pi\right) \star \left(\sum_{\pi \in S_n} b_\pi \pi\right) = \sum_{\pi, \pi' \in S_n} a_\pi b_{\pi'} \pi \cdot \pi'$.

Irreducible representations of $S_n$ can be labeled with partitions of $n$ (e.g. $\{2, 1, 1\}$ is a partition of $n = 4$ since $2 + 1 + 1 = 4$) which in turn are often depicted graphically as Young diagrams. In the case of $n = 3$, there are three irreducible representations: $\mathbf{1}$(the trivial one), $\mathbf{1'}$ (the alternating one) and $\mathbf{2}$. They are associated to the partitions $\{3\}$, $\{1, 1, 1\}$ and $\{2, 1\}$, i.e.

$$\square\square\square \, , \quad \begin{matrix}\square\\\square\\\square\end{matrix} \text{ and } \begin{matrix}\square\square\\\square\end{matrix} . \tag{36}$$

## Young symmetrizers

> **`YoungSymmetrizeTensor[<tensor>,<Young tableaux>]`**
>
> Returns the given tensor projected with the Young symmetrizer associated to the second argument.

With this function one can symmetrize a tensor according to some tableaux $\lambda$. The symmetrizer associated to a Young tableaux can be defined as follows. Consider the sets of elements of the permutation group $S_n$ which leave the rows and columns of $\lambda$ invariant:

$$H_\lambda = \{\pi \in S_n : \pi \text{ does not change the rows of } \lambda\} , \tag{37}$$

$$V_\lambda = \{\pi \in S_n : \pi \text{ does not change the columns of } \lambda\} . \tag{38}$$

Then the Young symmetrizer associated to $\lambda$ is is taken to be

$$y_\lambda \equiv a_\lambda s_\lambda \text{ with } s_\lambda = \sum_{h \in H_\lambda} h \text{ and } a_\lambda = \sum_{v \in V_\lambda} \text{sign}\,(v)\,v . \tag{39}$$

For an input tensor $T$, the function `YoungSymmetrizeTensor` returns $y_\lambda T$, reusing the index labels given in the input. Each tableaux should be provided as lists; for example

$$\begin{matrix}\boxed{1}\boxed{3}\\\boxed{2}\end{matrix} = \{\{1, 3\}, \{2\}\}, \quad \boxed{1}\boxed{2} = \{\{1, 2\}\} . \tag{40}$$

```
In[ ]:= YoungSymmetrizeTensor[Y[p, q, r], {{1, 3}, {2}}]
```

$$Out[ ]= \frac{1}{3} Y[p, q, r] - \frac{1}{3} Y[q, p, r] - \frac{1}{3} Y[q, r, p] + \frac{1}{3} Y[r, q, p]$$

```
In[ ]:= YoungSymmetrizeTensor[S[x1, x2], {{1, 2}}]
```

$$Out[ ]= \frac{1}{2} S[x1, x2] + \frac{1}{2} S[x2, x1]$$

## $S_n$ **irreps in a tensor**

```
SnIrrepsInTensor[<null conditions encoding the tensor symmetries>]
```

Returns the non-null components of a tensor with the given symmetry. The output is a list of the form {{partition1, multiplicity1}, ...}.

Consider a general rank-$n$ tensor, with no symmetries, where all indices are of the same nature. Assuming that each index can take $m$ values, one can split the $m^n$ components of $T$ according to how they transform under $S_n$ permutations $T_{i_1 \cdots i_n} \to T_{i_{\pi(1)} \cdots i_{\pi(n)}}$. For an $S_n$ irreducible representation labeled by a partition $\lambda$ of $n$ there are precisely $d(\lambda)$ parts of $T$ transforming as $\lambda$, where $d(\lambda)$ is the dimension of the irreducible representation. For example a 3-index tensor can be split in 4 parts, each transforming according to the following irreps:

$$\square\square\square + \begin{array}{c}\square\square\\\square\end{array} + \begin{array}{c}\square\square\\\square\end{array} + \begin{array}{c}\square\\\square\\\square\end{array} \tag{41}$$

One can use Young symmetrizers to project out each of them. Out of a total of $m^n$, the number of components associated to each piece can be computed from $m$ and the shape of each diagram $\lambda$, using a well known formula which is not important for the present discussion; the interested reader can find more details in [19] (see also the HookContentFormula function in [16]).

Importantly, in a tensor with symmetries (see appendix B) some of these parts are constrained. They could be zero, or perhaps have relations among themselves; in either case, they are not all independent. For instance, in a rank-3 fully symmetry tensor there is only the $\square\square\square$ piece.

Based on a tensor's symmetry equations, the function SnIrrepsInTensor computes the non-zero components. Here are two examples:

```
In[ ]:= SnIrrepsInTensor[{R[f2, f1, f3, f4] + R[f1, f2, f3, f4],
        R[f1, f2, f3, f4] + R[f1, f2, f4, f3], R[f1, f2, f3, f4] - R[f3, f4, f1, f2],
        R[f1, f2, f3, f4] + R[f1, f3, f4, f2] + R[f1, f4, f2, f3]}]
```

$$Out[ ]= \left\{\left\{\begin{array}{cc}\square&\square\\\square&\square\end{array}, 1\right\}\right\}$$

```
In[ ]:= SnIrrepsInTensor[{YoungSymmetrizeTensor[Y[p, q, r], {{1, 2}, {3}}]}]
```

$$Out[ ]= \left\{\{\square\square\square, 1\}, \left\{\begin{array}{c}\square\square\\\square\end{array}, 1\right\}, \left\{\begin{array}{c}\square\\\square\\\square\end{array}, 1\right\}\right\}$$

Note that the second case corresponds, for $\lambda = \{\{1,2\}, \{3\}\}$, to $y_\lambda Y$ set to zero (not $y_\lambda Y = Y$).

The program will consider that all tensor indices are of the same nature, always decomposing a rank-$n$ tensor in $S_n$ irreps. Sometimes that might not the best approach: take for example a tensor $P$ with the symmetry $P_{abcd} = P_{badc}$. Even though the first two indices never swap with the last two, and therefore one could consider just the small permutation group $S_2 \times S_2$, currently the function `SnIrrepsInTensor` ignores this and decomposes the tensor in $S_4$ irreps:

$In[\circ]:=$ **SnIrrepsInTensor[{P[a, b, c, d] - P[b, a, d, c]}]**

$Out[\circ]=$ $\left\{ \{\square\square\square\square, 1\}, \left\{ \begin{array}{c}\square\square\square\\\square\end{array}, 1\right\}, \{\boxplus, 2\}, \left\{\begin{array}{c}\square\square\\\square\\\square\end{array}, 1\right\}, \left\{\begin{array}{c}\square\\\square\\\square\\\square\end{array}, 1\right\} \right\}$

## Condensing various tensor symmetry equations into a single projector

**SingleProjector[<null conditions with the tensor symmetries>]**

Returns the unique hermitian projector P such that the condition P(tensor)=tensor is equivalent to the set of null equations given as input.

It might sometimes be convenient to reduce a set of symmetry equations into a single one. The `SingleProjector` function does that, by returning a single hermitian operator $P$, which contains all the input symmetries.[6] Requiring that $P$ is a projector ($P^2 = P$) and hermitian ($P^\dagger = P$) makes it unique.[7]

---

[6]Note that the input, as always, must be a list $\{\text{expr1}, \text{expr2}, ...\}$ of null expressions, i.e. $\text{expr1} = \text{expr2} = \cdots = 0$, while the output is a projector $P$ such that $P(\text{tensor}) = \text{tensor}$. So $(P - e)(\text{tensor})$ is a single null expression equivalent to the original list.

[7]The adjoint can be understood as follows. For two tensors $A$ and $B$ of the same rank, one can define the inner product $\langle A, B \rangle = A^*_{i_1 i_2 \cdots i_n} B_{i_1 i_2 \cdots i_n}$. Then, for some member $\mathcal{X} = \sum_{\pi \in S_n} c_\pi \pi$ of the $S_n$ algebra, where the $c_\pi$ are complex numbers,

$$\langle A, \mathcal{X}B \rangle \equiv \langle \mathcal{X}^\dagger A, B \rangle = \sum_{\pi \in S_n} c_\pi A^*_{i_1 i_2 \cdots i_n} B_{\pi(i_1 i_2 \cdots i_n)}$$

which is the same as $\sum_{\pi \in S_n} c_\pi A^*_{\pi^{-1}(i_1 i_2 \cdots i_n)} B_{i_1 i_2 \cdots i_n}$ or simply $\sum_{\pi \in S_n} \left[ c^*_{\pi^{-1}} A_{\pi(i_1 i_2 \cdots i_n)} \right]^* B_{i_1 i_2 \cdots i_n}$. As such

$$\mathcal{X}^\dagger = \sum_{\pi \in S_n} c^*_{\pi^{-1}} \pi \,.$$

For rank-2 tensors, it is well know that if $A$ is (anti)symmetric then only the (anti)symmetric part of $B$ contributes to the contraction $A_{ij} B_{ij}$. We are thus entitled to think of $A$ and $B$ as having the exact same symmetry. However, for higher rank indices this is only true if $\mathcal{X} = \mathcal{X}^\dagger$.

```
In[ ]:= SinglePProjector[{R[f2, f1, f3, f4] + R[f1, f2, f3, f4],
         R[f1, f2, f3, f4] + R[f1, f2, f4, f3], R[f1, f2, f3, f4] - R[f3, f4, f1, f2],
         R[f1, f2, f3, f4] + R[f1, f3, f4, f2] + R[f1, f4, f2, f3]}]
```

$$Out[\cdot]= \frac{1}{12} R[f1, f2, f3, f4] - \frac{1}{12} R[f1, f2, f4, f3] + \frac{1}{24} R[f1, f3, f2, f4] - \frac{1}{24} R[f1, f3, f4, f2] -$$

$$\frac{1}{24} R[f1, f4, f2, f3] + \frac{1}{24} R[f1, f4, f3, f2] - \frac{1}{12} R[f2, f1, f3, f4] + \frac{1}{12} R[f2, f1, f4, f3] -$$

$$\frac{1}{24} R[f2, f3, f1, f4] + \frac{1}{24} R[f2, f3, f4, f1] + \frac{1}{24} R[f2, f4, f1, f3] - \frac{1}{24} R[f2, f4, f3, f1] -$$

$$\frac{1}{24} R[f3, f1, f2, f4] + \frac{1}{24} R[f3, f1, f4, f2] + \frac{1}{24} R[f3, f2, f1, f4] - \frac{1}{24} R[f3, f2, f4, f1] +$$

$$\frac{1}{12} R[f3, f4, f1, f2] - \frac{1}{12} R[f3, f4, f2, f1] + \frac{1}{24} R[f4, f1, f2, f3] - \frac{1}{24} R[f4, f1, f3, f2] -$$

$$\frac{1}{24} R[f4, f2, f1, f3] + \frac{1}{24} R[f4, f2, f3, f1] - \frac{1}{12} R[f4, f3, f1, f2] + \frac{1}{12} R[f4, f3, f2, f1]$$

Note that Young symmetrizers are in general not hermitian (see for example [20] and references contained therein):

```
In[ ]:= YoungSymmetrizeTensor[Y[p, q, r], {{1, 3}, {2}}]
         SingleProjector[{Y[p, q, r] - %}]
```

$$Out[\cdot]= \frac{1}{3} Y[p, q, r] - \frac{1}{3} Y[q, p, r] - \frac{1}{3} Y[q, r, p] + \frac{1}{3} Y[r, q, p]$$

$$Out[\cdot]= \frac{1}{3} Y[p, q, r] - \frac{1}{6} Y[p, r, q] - \frac{1}{6} Y[q, p, r] - \frac{1}{6} Y[q, r, p] - \frac{1}{6} Y[r, p, q] + \frac{1}{3} Y[r, q, p]$$

## Comparing sets of symmetry relations

`SameEquationsQ[<null equations 1>, <null equations 2>]`

Compares the two sets of symmetry conditions (which may contain one or more tensor heads). The output is a string which identifies one out of 4 possible cases: (a) the equations are identical; (b) equations #1 are more restrictive than equations #2; (c) equations #2 are more restrictive than equations #1; (d) none of these apply (equations #1 and #2 are different).

It might be important to know if two sets of equations, involving tensors and their permutations, are the same or not. This can be checked with the function `SameEquationsQ`. Note that the equations can contain one of more tensors.

As a very simple example, the equation

$$P_{k_1 k_2} = Q_{k_1 k_2} \tag{42}$$

is equivalent to the following two equations:

$$P_{k_1 k_2} + P_{k_2 k_1} = Q_{k_1 k_2} + Q_{k_2 k_1} \text{ and } P_{k_1 k_2} - P_{k_2 k_1} = Q_{k_1 k_2} - Q_{k_2 k_1}. \tag{43}$$

On the other hand, (42) is more restrictive than

$$P_{k_1 k_2} + P_{k_2 k_1} = Q_{k_1 k_2} + Q_{k_2 k_1} \tag{44}$$

which only forces the symmetric part of the two tensors to be the same. Finally

$$P_{k_1 k_2} = Q_{k_2 k_1} \tag{45}$$

in neither equal, nor more restrictive, nor included in condition (42), so in this sense one can say that is altogether different/unrelated to (42). These comparisons can be performed with the following code:

```
eqs1 = {P[k1, k2] - Q[k1, k2]};
eqs2 = {P[k1, k2] + P[k2, k1] - (Q[k1, k2] + Q[k2, k1]),
    P[k1, k2] - P[k2, k1] - (Q[k1, k2] - Q[k2, k1])};
eqs3 = {P[k1, k2] + P[k2, k1] - (Q[k1, k2] + Q[k2, k1])};
eqs4 = {P[k1, k2] - Q[k2, k1]};

SameEquationsQ[eqs1, eqs2]
SameEquationsQ[eqs1, eqs3]
SameEquationsQ[eqs1, eqs4]
```

*Out[•]=* Same system of equations

*Out[•]=* Equations #1 are more restrictive

*Out[•]=* Equations #1 and #2 are different

As a further example, one can check that $R_{abcd} = -R_{abdc}$ and $-R_{bacd} + R_{acdb} + R_{adbc} = 0$ are the same as the set of equations in (1) for the Riemann tensor:

```
In[•]:= RiemannSyms1 = {R[f2, f1, f3, f4] + R[f1, f2, f3, f4],
    R[f1, f2, f3, f4] + R[f1, f2, f4, f3], R[f1, f2, f3, f4] - R[f3, f4, f1, f2],
    R[f1, f2, f3, f4] + R[f1, f3, f4, f2] + R[f1, f4, f2, f3]};
RiemannSyms2 = {R[f1, f2, f3, f4] + R[f1, f2, f4, f3],
    -R[f2, f1, f3, f4] + R[f1, f3, f4, f2] + R[f1, f4, f2, f3]};
SameEquationsQ[RiemannSyms1, RiemannSyms2]
```

*Out[•]=* Same system of equations

In fact, with **SingleProjector** one can express these symmetries as a single equation:

```
In[•]:= SingleProjector[RiemannSyms1]
       SameEquationsQ[RiemannSyms1, {R[f1, f2, f3, f4] - %}]
```

$$Out[•]= \frac{1}{12} R[f1, f2, f3, f4] - \frac{1}{12} R[f1, f2, f4, f3] + \frac{1}{24} R[f1, f3, f2, f4] - \frac{1}{24} R[f1, f3, f4, f2] -$$

$$\frac{1}{24} R[f1, f4, f2, f3] + \frac{1}{24} R[f1, f4, f3, f2] - \frac{1}{12} R[f2, f1, f3, f4] + \frac{1}{12} R[f2, f1, f4, f3] -$$

$$\frac{1}{24} R[f2, f3, f1, f4] + \frac{1}{24} R[f2, f3, f4, f1] + \frac{1}{24} R[f2, f4, f1, f3] - \frac{1}{24} R[f2, f4, f3, f1] -$$

$$\frac{1}{24} R[f3, f1, f2, f4] + \frac{1}{24} R[f3, f1, f4, f2] + \frac{1}{24} R[f3, f2, f1, f4] - \frac{1}{24} R[f3, f2, f4, f1] +$$

$$\frac{1}{12} R[f3, f4, f1, f2] - \frac{1}{12} R[f3, f4, f2, f1] + \frac{1}{24} R[f4, f1, f2, f3] - \frac{1}{24} R[f4, f1, f3, f2] -$$

$$\frac{1}{24} R[f4, f2, f1, f3] + \frac{1}{24} R[f4, f2, f3, f1] - \frac{1}{12} R[f4, f3, f1, f2] + \frac{1}{12} R[f4, f3, f2, f1]$$

```
Out[•]= Same system of equations
```

## Symmetries of a numerical tensor

**SymmetriesOfNumericalTensor[<numerical tensor>]**

Returns a list of null equations encoding the symmetries of the given tensor, whose components are named "tensor"[id1,id2,...] in the output.

One might want to know the symmetries of a known tensor, whose entries are just numbers. The function SymmetriesOfNumericalTensor returns a list of null expressions, which taken together contain all the symmetry information of a given numerical tensor. Rather than a single long expression, the result is often a list of several short null conditions applicable to the tensor (if desired, these can be condensed into a single one with the help to the SingleProjector function). Here are some examples:

```
In[•]:= SymmetriesOfNumericalTensor[LeviCivitaTensor[3]]

Out[•]= {tensor[id1, id2, id3] + tensor[id3, id2, id1],
         tensor[id1, id3, id2] - tensor[id3, id2, id1]}

In[•]:= SymmetriesOfNumericalTensor[TensorProduct[LeviCivitaTensor[2], LeviCivitaTensor[2]]]

Out[•]= {tensor[id1, id2, id3, id4] - tensor[id4, id3, id2, id1],
         tensor[id1, id2, id4, id3] + tensor[id4, id3, id2, id1],
         tensor[id1, id4, id2, id3] - tensor[id4, id2, id3, id1] + tensor[id4, id3, id2, id1]}

In[•]:= SymmetriesOfNumericalTensor[TensorProduct[IdentityMatrix[7], IdentityMatrix[7]]]

Out[•]= {tensor[id1, id2, id3, id4] - tensor[id4, id3, id2, id1],
         tensor[id1, id2, id4, id3] - tensor[id4, id3, id2, id1]}
```

# B. Tensor symmetries beyond Young symmetrizers

There is a widespread understanding that the symmetry of a tensor can always be described with Young symmetrizers. In this appendix I will argue why this is not the case. The following discussion is generalizable to any tensor, but for concreteness and simplicity let us consider a 3-index tensor $T_{ijk}$, where each index takes $m$ values. We want to consider all symmetries that $T$ may possibly have. Such symmetries can be expressed as

$$P\left(T_{ijk}\right) = T_{ijk} \tag{46}$$

or equivalently $(P - e)\left(T_{ijk}\right) = 0$, with $P = P^2$ being some member of the 6-dimensional algebra of $S_3$:

$$P = x_1 e + x_2\,(12) + x_3\,(13) + x_4\,(23) + x_5\,(123) + x_6\,(132) \ . \tag{47}$$

The symmetries of a tensor might be given as a list of several (rather than just one) equations of this type. However, one can always translate those constraints into a single equation of the form (46) (and indeed this can be achieved with the function `SingleProjector` presented in this work).

Returning to the $T_{ijk}$ tensor, its $m^3$ components can then be split in the following parts:

- A symmetric part $T_a^{\square\square\square}$ with $\frac{1}{6}m\,(m+1)\,(m+2)$ components. It can be projected out with the Young symmetrizer $Y_{\square\square\square} = \frac{1}{6}\left[e + (12) + (13) + (23) + (123) + (132)\right]$.

- An antisymmetric part $T_a^{\square}$ with $\frac{1}{6}m\,(m-1)\,(m-2)$ components. It can be projected out with the Young symmetrizer $Y_{\square} = \frac{1}{6}\left[e - (12) - (13) - (23) + (123) + (132)\right]$.

- A part with mixed symmetry $\boxplus$, having a total of $\frac{2}{3}\,(m+1)\,m\,(m-1)$ components. It can be projected out by the sum of Young symmetrizers $Y_{\boxplus}^{(1)} + Y_{\boxplus}^{(2)}$ with $Y_{\boxplus}^{(1,2)}$ given below in equations (49) and (50).

Since $\boxplus$ is a 2-dimensional representation of the permutation group, one can further split the space of mixed-symmetry components in two, each with $\frac{1}{3}\,(m+1)\,m\,(m-1)$ components: $T_{i,a}^{\boxplus}$ with $i = 1, 2$ and $a = 1, \cdots, \frac{1}{3}\,(m+1)\,m\,(m-1)$. Combinations of the six permutations act on the first index ($i$) only. In fact, one can make arbitrary transformations in this 2-dimensional space spanned by the $i$ index, meaning that with a suitable combination $P$ of the form (47) applied to the tensor $T$ one can achieve any linear transformation

$$\begin{pmatrix} T_{1,a}^{\boxplus} \\ T_{2,a}^{\boxplus} \end{pmatrix} \rightarrow \overbrace{\begin{pmatrix} M_{11} & M_{12} \\ M_{21} & M_{22} \end{pmatrix}}^{M} \cdot \begin{pmatrix} T_{1,a}^{\boxplus} \\ T_{2,a}^{\boxplus} \end{pmatrix} \ . \tag{48}$$

Note that each of the 4 $M_{ij}$'s is associated with an element (47) of the algebra of $S_3$; together with $Y_{\square\square\square}$ and $Y_{\square}$ they form a basis for this 6-dimensional algebra.

There is an arbitrariness in defining a basis for the 2-dimension space $\left\{T_{1,a}^{\boxplus}, T_{2,a}^{\boxplus}\right\}$; a convenient way of fixing it is with Young symmetrizers, such as

$$Y_{\boxplus}^{(1)} = \frac{1}{3}\left[e + (12)\right]\left[e - (13)\right] = \frac{1}{3}\left[e + (12) - (13) - (132)\right] \ , \tag{49}$$

$$Y^{(2)}_{\yng(2,1)} = \frac{1}{3}\left[e + (13)\right]\left[e - (12)\right] = \frac{1}{3}\left[e + (13) - (12) - (123)\right] . \tag{50}$$

One can then define $T^{\yng(2,1)}_{i,a}$ to be such that[8]

$$Y^{(1)}_{\yng(2,1)}\begin{pmatrix} T^{\yng(2,1)}_{1,a} \\ T^{\yng(2,1)}_{2,a} \end{pmatrix} = \begin{pmatrix} 1 & 0 \\ 0 & 0 \end{pmatrix}\cdot\begin{pmatrix} T^{\yng(2,1)}_{1,a} \\ T^{\yng(2,1)}_{2,a} \end{pmatrix} \text{ and } Y^{(2)}_{\yng(2,1)}\begin{pmatrix} T^{\yng(2,1)}_{1,a} \\ T^{\yng(2,1)}_{2,a} \end{pmatrix} = \begin{pmatrix} 0 & 0 \\ 0 & 1 \end{pmatrix}\cdot\begin{pmatrix} T^{\yng(2,1)}_{1,a} \\ T^{\yng(2,1)}_{2,a} \end{pmatrix} . \tag{51}$$

Therefore, a $Y^{(1)}_{\yng(2,1)}$-symmetric tensor $T$, $T_{ijk} = Y^{(1)}_{\yng(2,1)}(T_{ijk})$, will not have the $T^{\yng(2,1)}_{2,a}$ part (nor the $T^{\yng(3)}_{a}$ parts). On the other hand, if $T_{ijk} = Y^{(2)}_{\yng(2,1)}(T_{ijk})$ the tensor $T$ will have $T^{\yng(2,1)}_{1,a}$ zeroed out. Clearly in both the cases the tensor is no longer general but rather has a special, symmetric form.

The crucial point is that one can achieve arbitrary matrices $M$ in equation (48), including off-diagonal ones, so there are symmetric tensors which cannot be described only with the projectors $Y^{(1,2)}_{\yng(2,1)}$. To make this observation more concrete, first note that the transformation in (48) can be achieved with the following element of the $S_3$ algebra:

$$P_M \equiv \frac{M_{11} + M_{22}}{3}e + \frac{M_{11} + M_{12} - M_{22}}{3}(12) + \frac{-M_{11} + M_{21} + M_{22}}{3}(13) + \frac{-M_{12} - M_{21}}{3}(23)$$
$$+ \frac{M_{12} - M_{21} - M_{22}}{3}(123) + \frac{-M_{11} - M_{12} + M_{21}}{3}(132) . \tag{52}$$

For any two matrices $M$ and $N$, one can check that, as it should, $P_M P_N = P_{MN}$. Furthermore $Y_{\yng(3)}P_M = Y_{\yng(1,1,1)}P_M = 0$.

For completeness, as a final step let us find all 2 by 2 matrices $M$ which are projectors, as we want to apply $P_M$ to the $T$ tensor: $P_M(T_{ijk}) = T_{ijk}$. One trivial possibility is $M = \mathbf{0}$ which zeroes out the full $\yng(2,1)$ space of the $T$ tensor. The other trivial possibility is $M = \mathbf{1}$, which keeps all this space; note that $P_{\mathbf{1}}$ can be expressed using the Young projectors: $P_{\mathbf{1}} = Y^{(1)}_{\yng(2,1)} + Y^{(2)}_{\yng(2,1)}$.

The more interesting scenario for the present discussion is when the eigenvalues of $M$ are non-degenerate. We pick an eigenvector $(\cos\alpha, \sin\alpha)^T$ for the eigenvalue 1 and another $(-\sin\beta, \cos\beta)^T$ for the eigenvalue 0, yielding the matrix

$$M = \frac{1}{\cos(\alpha - \beta)}\begin{pmatrix} \cos\alpha\cos\beta & \cos\alpha\sin\beta \\ \sin\alpha\cos\beta & \sin\alpha\sin\beta \end{pmatrix} . \tag{53}$$

For this particular $M$,

$$P_M(T_{ijk}) = T_{ijk} \Rightarrow \begin{pmatrix} T^{\yng(2,1)}_{1,a} \\ T^{\yng(2,1)}_{2,a} \end{pmatrix} \propto \begin{pmatrix} \cos\alpha \\ \sin\alpha \end{pmatrix} \text{ i.e. } T^{\yng(2,1)}_{2,a} = \tan\alpha\, T^{\yng(2,1)}_{1,a} . \tag{54}$$

Note that while the value of $P_M$ changes with $\beta$, this angle does not impact the tensor $T$ itself. Also notice that $P_M$ does not really need to be a projector: $P_M(T_{ijk}) = T_{ijk}$ will kill any component of $T$ which is not associated to an eigenvalue 1, hence the exact value of the other eigenvalues

---

[8]Actually the two Young projectors only define $\left\{T^{\yng(2,1)}_{1,\alpha}, T^{\yng(2,1)}_{2,\alpha}\right\}$ up to (two) multiplicative factors.

$\neq 1$ are irrelevant. In fact, the only thing that matters are the eigenvectors of $P_M$ associated the eigenvalue 1.

But the most important observation is that one can have symmetric tensors whose symmetry cannot be expressed with Young projectors. The latter can be used to set $T_{1,a}^{\tiny\yng(2)} = 0$ or $T_{2,a}^{\tiny\yng(1,1)} = 0$, but it is clear that one can have arbitrary relations between $T_{1,a}^{\tiny\yng(2)}$ and $T_{2,a}^{\tiny\yng(1,1)}$.

The discussion above holds true for tensors with higher rank, and indeed the insufficiency of Young symmetrizers becomes more acute as we consider larger irreducible representations of the permutation group. For example, if $T$ has four indices, we can split it into the 10 parts indicated below, and the elements of the 24-dimensional algebra of $S_4$ act on it as follows:

$$
\begin{pmatrix} T_a^{\tiny\yng(4)} \\ T_{1,a}^{\tiny\yng(3,1)} \\ T_{2,a}^{\tiny\yng(3,1)} \\ T_{3,a}^{\tiny\yng(3,1)} \\ T_{1,a}^{\tiny\yng(2,2)} \\ T_{2,a}^{\tiny\yng(2,2)} \\ T_{1,a}^{\tiny\yng(2,1,1)} \\ T_{2,a}^{\tiny\yng(2,1,1)} \\ T_{3,a}^{\tiny\yng(2,1,1)} \\ T_a^{\tiny\yng(1,1,1,1)} \end{pmatrix}
\rightarrow
\begin{pmatrix}
\times & & & & & & & & & \\
& \times & \times & \times & & & & & & \\
& \times & \times & \times & & & & & & \\
& \times & \times & \times & & & & & & \\
& & & & \times & \times & & & & \\
& & & & \times & \times & & & & \\
& & & & & & \times & \times & \times & \\
& & & & & & \times & \times & \times & \\
& & & & & & \times & \times & \times & \\
& & & & & & & & & \times
\end{pmatrix}
\cdot
\begin{pmatrix} T_a^{\tiny\yng(4)} \\ T_{1,a}^{\tiny\yng(3,1)} \\ T_{2,a}^{\tiny\yng(3,1)} \\ T_{3,a}^{\tiny\yng(3,1)} \\ T_{1,a}^{\tiny\yng(2,2)} \\ T_{2,a}^{\tiny\yng(2,2)} \\ T_{1,a}^{\tiny\yng(2,1,1)} \\ T_{2,a}^{\tiny\yng(2,1,1)} \\ T_{3,a}^{\tiny\yng(2,1,1)} \\ T_a^{\tiny\yng(1,1,1,1)} \end{pmatrix},
\tag{55}
$$

where each cross can take any value (note that there are precisely $24 = 4!$ of them). If, for example, we look at the 3-dimensional $\tiny\yng(3,1)$ subspace, we find that the three Young symmetrizers are represented by the matrices

$$
\begin{pmatrix} 1 & 0 & 0 \\ 0 & 0 & 0 \\ 0 & 0 & 0 \end{pmatrix}, \begin{pmatrix} 0 & 0 & 0 \\ 0 & 1 & 0 \\ 0 & 0 & 0 \end{pmatrix}, \begin{pmatrix} 0 & 0 & 0 \\ 0 & 0 & 0 \\ 0 & 0 & 1 \end{pmatrix}
\tag{56}
$$

and again they are insufficient to express every conceivable tensor symmetry. Take for instance the symmetry $T_{1,a}^{\tiny\yng(3,1)} = T_{2,a}^{\tiny\yng(3,1)} + 2T_{3,a}^{\tiny\yng(3,1)}$: none of the $T_{i,a}^{\tiny\yng(3,1)}$ is null so this kind of relation cannot be obtained with Young symmetrizers, and yet it can be achieved with the matrix

$$
M = \begin{pmatrix} 1 & 0 & 0 \\ 0 & 1 & 0 \\ \frac{1}{2} & -\frac{1}{2} & 0 \end{pmatrix}
\tag{57}
$$

since it has eigenvectors $(1,1,0)^T$ and $(2,0,1)^T$ associated to the eigenvalue 1.

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
