# Peer review of "Using SimTeEx to simplify polynomial expressions with tensors"

_SciPost Physics Codebases_

## Round 2 · Referee Report · Anonymous (Referee 1) · 2025-3-27

Strengths

1 - There are some well-explained examples in the algorithm's presentation.

Weaknesses

1 - Limited Novelty of the Approach

2 - Absence of Comparison with Existing Tools and Algorithms

3 - Lack of Clarity, Particularly in the Initial Sections

4 - Informal Tone and Redundancy in the Writing

Report

The paper introduces SimTeEx, a Mathematica package designed to simplify polynomial expressions involving tensors with symmetries. The author claims that the method generalizes to any tensor symmetry and provides examples demonstrating its application.

**Overall evaluation:**
Although the problem statement is important and the ideas are sound, the paper requires significant revisions (see below). Also, the paper does not meet the acceptence critera for Scipost Physics Codebase since benchmarks are missing.
Although this might be possible, I am uncertain if this can be managed to be done in a certain time-frame.
These revisions must at least include
(*) Adding benchmarks with existing software
(*) Proving novelty: by comparing the usability with existing software and showing which instances can only be treated by the new approach.
(*) Relating the mathematical background (especially Section 2-3) to the tools presented in the rest of the paper.

**More details on the weaknesses:**
As noted in the Weaknesses section, several aspects of the paper require substantial revision before it can be considered for acceptance.

1 - Limited Novelty of the Approach
While the problem of simplifying tensor expressions is relevant, it has already been addressed in numerous previous works. Presenting new algorithms for this problem is crucial. However, the author states (for example on pages 2-3 and on many other places of the manuscript) that the presented algorithm shares many similarities with existing packages such as Redberry, ATENSOR, Cadabra, and xPerm.

From the manuscript as currently written, it is unclear in what way the methods presented are novel. The author briefly mentions that certain instances can be handled by SimTeEx that one other tool cannot (e.g., at the end of page 11), but the manuscript lacks a thorough and detailed discussion of the specific types of expressions and symmetries that SimTeEx can handle, which are not addressed by existing tools. An in-depth comparison of the capabilities of SimTeEx and other packages would be helpful, particularly outlining the cases where SimTeEx demonstrates clear advantages.

2 - Comparison with Other Tools with respect to Performance Benchmarks:
There is no direct comparison in the paper regarding the complexity of the current tool relative to others. A comparison with existing tools in terms of performance and computational efficiency is essential. It would be valuable to include benchmarks demonstrating how SimTeEx performs relative to packages like Redberry, ATENSOR, Cadabra, or xPerm, both in terms of time complexity and handling more complex tensor expressions.

This is also in conflict with one of the acceptence criteria for this journal: "Benchmarking tests must be provided."

3 - Lack of Coherence and Structure
The manuscript lacks a consistent, cohesive narrative. For example, the connection between Section 2 (dummy indices), Section 3 (polynomials as a vector space) and the rest of the paper is unclear. It is important for the paper to clearly explain how these sections contribute to the overall methodology and how they relate to the implementation of SimTeEx. Without this explanation, the paper seems disjointed, and the relevance of certain sections remains ambiguous.

4 - Informal Tone and Redundancy in the Writing
In several places, the manuscript could benefit from a more formal and concise writing style to enhance clarity and professionalism. There are also multiple grammatical/typographical errors in the manuscript. For example, in the Introduction:

"I have tried to make the discussion there somewhat self-contained, however textbooks on the matter, such as [10], might still come in handy." (page 2)

This phrasing is quite informal and could be revised to present the core ideas more directly without referring to external sources for introductory material.

"Having said this, the reader unfamiliar with the permutation group and its representations shouldn’t be overly concerned as the algorithm described in this paper does not rely on it, nor does one need to know any of this in order to use the main function of the SimTeEx program (to be introduced later), which puts a tensor expression in canonical form." (page 2)

The sentence is lengthy and could be streamlined for improved readability.

Additionally, there are several grammatical issues and typographical errors, which are noted in the requested changes below. The manuscript also occasionally uses the future tense where a different tense would be more appropriate, for example:

"On this topic, I will note that the user is free to [..]"

A more consistent and precise writing style would strengthen the overall presentation of the paper.

Requested changes

In order to decide whether the paper can be accepted the author needs to add several additional information:

(1) A clear information what can/cannot be done by this algorithm compared to other packages such as Redberry, ATENSOR, Cadabra, and xPerm

(2) A benchmark test, comparing the tool with the other packages

(3) Setting a connection between Section 2, 3 and the rest of the paper.

Some of the grammatical and typographical errors found:

"Likewise, under the assumption that k is also fully symmetry [...]" -> is also fully symmetric

well known -> well-known

"Things becomes more complicated [...]" -> become more complicated

"The user does not need to know what is the representation" -> what the representation is

"In fact the user is free to provide" -> In fact, the user is [...]

"code supporting multi-term symmetries which I am aware of" -> that I am aware of

"One should be aware that the function CanonicalForm when applied to two equivalent expressions," -> CanonicalForm, when

"although it is always be true that" -> "although it is always true that"

"performs the sorting operation which is mentioned at the end of section 4." -> performs the sorting operation mentioned at the end of section 4.

"not to introduce new ones which are unfamiliar to the user." -> ones that are unfamiliar

"On often encounters" -> one

"simple modifications to algorithm presented" -> to the algorithm presented

"describe an algorithm which can take into account" -> that can take into account

Recommendation

Ask for major revision

  • validity: ok
  • significance: high
  • originality: high
  • clarity: ok
  • formatting: excellent
  • grammar: reasonable

Author:  Renato Fonseca  on 2025-06-30  [id 5607]

(in reply to Report 1 on 2025-03-27)
Category:
remark
answer to question
reply to objection
correction

Dear editor, I appreciate the opportunity to respond to the referee report.

Let me start by expressing concern for some of the unfounded criticisms in the report, as well as for the placement of undue importance on matters of writing style. I invite you and the witnesses of this review process to see for themselves that the claims of 'lack of clarity', 'informal tone', 'redundancy in the writing', and 'grammatical issues' are exaggerated at best. The 12 'grammatical issues' in the report, some of which are missing commas or hyphens, add up to less than one issue every two pages. I must therefore strongly disagree with the characterization of the grammar as being merely 'reasonable'.

With regard to the specific points raised in the report, I have made the following changes: - Grammatical and typographical errors pointed out by the referee (and for which I am thankful) were corrected. - Small changes of style were made to some sentences mentioned in the report. - Finally, a new appendix C was added with information on the performance of the code, as required by the journal. This new appendix also compares in more detail SimTeEx with other programs.

Below, I respond to several criticisms.

Concerning an alleged "Limited Novelty of the Approach", the report states

"While the problem of simplifying tensor expressions is relevant, it has already been addressed in numerous previous works. Presenting new algorithms for this problem is crucial. However, the author states (for example on pages 2-3 and on many other places of the manuscript) that the presented algorithm shares many similarities with existing packages such as Redberry, ATENSOR, Cadabra, and xPerm. (...)"

As mentioned in the text (pages 11, 12 and appendix B), and as far as I know, none of these packages can handle every possible tensor permutation. The new appendix C further highlights the differences among the codes. Note that neither xPerm/xTensor/xAct nor Redberry supports multi-term symmetries (see appendix C for a discussion of the implications of this). Furthermore, it does not seem possible to use ATENSOR to simplify even quadratic polynomials such as $R_{pqrs}R_{pqrs}+2R_{pqrs}R_{pqsr}$ where $R$ is the Riemann tensor. Cadabra does have a very interesting meld function; however, it seems to have the limitation described in the text (see pages 11, 12 and appendix B), namely a tensor's symmetry must be given by a Young tableau.

Also, I wouldn't call these other packages 'numerous' nor would I quantify the similarities as being 'many'.

I do mention in the manuscript, to credit earlier work, that some of the approaches used by SimTeEx (for example the ideas of using graphs and putting matrices in reduced row echelon form) have been mentioned previously in the literature in connection to simplifying tensor expressions. I learned about this after writing the code and while searching the literature, precisely to provide readers with potentially relevant references. Note that, for example, the tensor symmetries are linear relations so it is unsurprising that different authors arrive at the need of using, in one way or another, the row echelon form of some matrix in their codes.

An attentive reader, in good faith, will correctly infer that the similarities are quite vague: use of graphs; use of row echelon form. Diligently compiling relevant references and giving due credit to previous authors should not be regarded as a valid reason for criticism.

The report also states that

"The author claims that the method generalizes to any tensor symmetry ..."

I do make such a claim, and its truthfulness can be checked in two ways: (1) by inspecting the algorithm detailed in the paper and (2) by running the code. The sentence quoted above is never revisited in the report, hence I must lament the fact that the report leaves the suspicion that I, the author, made a false claim.

This is a central claim to the submission and obviously the novelty of the approach (which is the topic of the previous comment) can only be assessed by first confirming it.

The report claims that there is a "Lack of Coherence and Structure":

"The manuscript lacks a consistent, cohesive narrative. For example, the connection between Section 2 (dummy indices), Section 3 (polynomials as a vector space) and the rest of the paper is unclear. It is important for the paper to clearly explain how these sections contribute to the overall methodology and how they relate to the implementation of SimTeEx. Without this explanation, the paper seems disjointed, and the relevance of certain sections remains ambiguous."

I do not share this assessment. In my opinion, readers will find that the text has a consistent and cohesive narrative: - Tensor monomials and the problem of dummy indices are addressed in section 2 (using graphs). - Polynomials, i.e. sums of monomials, are discussed in section 3. -Tensor symmetries, which were ignored in the previous sections, are addressed in section 4.

I invite the witnesses of this review process to see how I framed this explanation at the end of the introduction (section 1), as well as in the initial parts of section 2, section 3 and section 4.

The report claims that there is an "Informal Tone and Redundancy in the Writing":

"In several places, the manuscript could benefit from a more formal and concise writing style to enhance clarity and professionalism. There are also multiple grammatical/typographical errors in the manuscript. ..."

Undue weight is placed on matters of writing style, while overlooking matters of substance, such as the novelty of the algorithm and its implementation in a code. As for the reference to enhancing professionalism, it risks being interpreted as personal rather than scholarly feedback, which I believe is best avoided in a peer review setting.

Three sentences are specifically mentioned in the report. I have made small changes to the first two while the third one ("On this topic, I will note that the user is free to [..]") was left unchanged.

Dear editor, to conclude, I have made some changes to the text and, crucially, I have added the benchmarks required by the journal. I hope that with these changes you and the rest of the Editorial College will find the manuscript ready for publication.

Attachment:

SimTeEx-resubmission.pdf

---

## Editorial Decision

resubmitted